# Analysis of the Influence of Pile-Raft Foundation Reinforcement on an Adjacent Existing Line Foundation

**Shen Zuo [1], Xianglong Zuo [1], Jin Li [1,*], Peng Liu [2,3,*] and Xinzhuang Cui [4]**

1   School of Civil Engineering, Shandong Jiaotong University, 5 Jiaoxiao Road, Jinan 250357, China
2   School of Civil Engineering, Central South University, 22 Shaoshan Road, Changsha 410075, China
3   National Engineering Laboratory for High Speed Railway Construction, Central South University, Changsha 410075, China
4   School of Civil Engineering, Shandong University, 27 Shanda Road, Jinan 250061, China
*   Correspondence: lijinsdjtu@163.com (J.L.); liupeng868@csu.edu.cn or lop868@163.com (P.L.)

**Abstract:** Pile-raft foundation reinforcement will have a certain impact on the adjacent existing line foundation, which will affect the normal service of the subgrade, and even lead to the instability of the subgrade. Until now, there have been few studies on adjacent construction problems, and there are few field experimental data available for us to consult. Therefore, this study relies on the construction project of Shanghai–Nanjing intercity high-speed railway close to the existing line, using in situ monitoring methods, such as stress shovels, horizontal strain gauges, and inclinometers combined with finite-element calculation and rail-inspection vehicle-data analysis. The stress, displacement, and geometric linearity of the adjacent existing line foundation during the reinforcement construction of a pile-raft foundation were studied. Our aim was to measure the impact and optimize the existing roadbed-protection measures employed during the construction period. We found that the cumulative horizontal displacement of the existing line foundation slope toe during the construction period was 24.25 mm, and the lateral displacement rate was less than 0.59 mm/d. The distance between the two lines was 9 m. The horizontal stress of the foundation soil in the depth range varied according to extrusion and retraction with the different construction stages, and the extrusion stress was less than 10 kPa. The horizontal stress changes in different construction stages were not obvious; the track quality index (TQI) in the existing track inspection during the construction period increased by 129.58%, and the existing track geometric linearity fluctuated greatly. According to the test results, the excavation stage of the subgrade foundation pit was the most dangerous stage of the existing subgrade during the construction of the new line pile-raft foundation. Although the change of the horizontal stress in different construction stages was not obvious, the horizontal displacement of the slope toe was sensitive to the construction process. Therefore, it could be used as a key indicator to monitor the deformation and stability of the existing subgrade. The correction coefficient was obtained by combining the detection data of the track-inspection vehicle with the finite element calculation data, based on which the accurate estimation of the horizontal displacement of the slope toe after the excavation of the foundation pit was realized. The monitoring and evaluation method of the stability of the existing line foundation under the influence of the pile-raft foundation reinforcement construction was preliminarily established through field monitoring and the analysis of the track-inspection data. Based on this method, relevant early warning values were proposed. The test results and engineering measures ensured the safe operation of the existing line foundation. This has important theoretical significance for guiding the construction of a new subgrade of adjacent existing lines.

**Keywords:** intercity railway; pile-raft foundation reinforcement; adjacent existing lines; field test; subgrade protection; track-inspection car

## 1. Introduction

The newly built Shanghai–Nanjing Intercity Railway and some sections of the existing Beijing–Shanghai line are adjacent to each other, with a parallel length of about 170 km and a distance of 4–15 m. The roadbed project adopted a new type of composite foundation with piles and rafts and construction processes, such as foundation excavation, pile formation, and filling. This will bring security risks to the adjacent existing railways.

The disturbance of structures surrounding new projects has increasingly attracted the attention of engineers and technicians, and it is now common to study the construction impacts of tunnels, foundation pits, and blasting [1,2]. Guo Dianta [3] used the superposition principle, the finite-element load-structure method, and the forced-displacement method. He calculated and analyzed the stability and deformation of the station structure adjacent to the foundation pit excavation. This study provided the control standard of station structure deformation for foundation pit construction. Ding Zhi [4–6] elaborated the research status of the construction problems adjacent to the soft soil foundation pit from four aspects: theoretical research, model testing, numerical simulation, and field analysis, and sorted and analyzed the measured data of different projects. It was found that there is a certain linkage between the deep displacement of the soil around the foundation pit, the deformation of the retaining structure, and the deformation of the adjacent subway tunnel. Gao Zikun [7] used theoretical formula derivation, which is based on solving the curvilinear equation of a quartic polynomial, considering the pile length, hole-wall shape, displacement boundary conditions, and soil material nonlinearity, and applied the variational principle to deduce the pile driving and extruding soil. The displacement, strain, and stress field solutions, and the research results, provide a theoretical basis for the selection and design of protective measures in the construction of pile pressing. Luo Zhanyou [8] researched the dynamic pile-pressing process using a finite-element model and field measurements. He calculated and simulated the vertical displacement field of soil extrusion caused by pile driving and compared the calculations with the actual measurements. The study found that the maximum compacting displacement delays the pile-sinking depth. Nan, L et al. [9] simulated the excavation process for the foundation pit through an indoor model test and obtained the deformation change and distribution law for the supporting structure. Zhang, P. [10] and others developed an artificial intelligence Internet of Things (AIoT)-based system for real-time monitoring of tunnel construction.

Foundation pit tunnels currently undergo routine deformation monitoring for supporting structures in accordance with regulations and specifications. With the development of detection technology, the use of radar monitoring methods such as geological advance forecasting to provide early warnings in the construction of underground caverns have received increasing attention in the engineering field [11–13]. The research results for the deformation control of structures such as tunnel foundation pits were relatively rich, covering the main technical means such as on-site monitoring, theoretical analysis, simulation calculations, and model tests, but less research has been carried out on the problem of subgrade deformation control in operation. Lan, Q. et al. [14] proposed a monitoring method for mountain highway slopes that used FEM analysis, a displacement meter, a rain gauge, and other hardware monitoring systems to test the displacement and rainfall. Jian-guang, L. [15] established a remote health monitoring system for mountain reservoirs and shore subgrades based on GPRS data transmission. Through the wireless remote transmission of safety monitoring data, the health status of subgrades was determined, which provided a basis for safe highway operation and subgrade disease prevention.

The existing monitoring of subgrade deformation is mainly used for new or special subgrades, and research on monitoring methods for adjacent construction affecting the operation subgrade remains rare. Rail-inspection vehicles are mainly used in China to detect the statuses of operating railway lines [16]. The "Railway Line Maintenance Rules" stipulate that the track quality index (TQI) should be used to evaluate the dynamic quality of the overall unevenness of the track-inspection vehicle. During the construction period, the existing roadbed is disturbed continuously, and there can be safety problems

at any time, but the detection frequency of the rail-inspection vehicle is far from meeting the requirements for the real-time monitoring of an existing roadbed during on-site construction.

In summary, the following problems exist in monitoring research on the impact of construction on an adjacent existing line foundation: (1) The rules and regulations do not specify a deformation alarm value for effects on structures such as railway subgrade engineering caused by temporary construction. (2) The adjacent existing railway lines must be completely closed for safety reasons, and thus, conventional roadbed monitoring and detection methods cannot be implemented. For this reason, this study considered the stress, strain, and distribution characteristics between the Shanghai–Nanjing Intercity Railway and the Beijing–Shanghai line in the three stages of construction. This study obtained key control indicators that can effectively reflect the impact of construction on the operation of subgrade safety through comparative analysis of track-inspection car data. This method ensured the safe operation of the existing line during the construction period. The safety monitoring method of the operating roadbed was also explored. In order to ensure the safe operation of the existing line during the construction of the pile-raft foundation reinforcement, on the basis of a static test, this paper carried out in situ monitoring experiments such as stress shovel, horizontal strain gauge, and inclinometer. Combined with the finite element calculation and the analysis of the track-inspection vehicle data, this research studies the stress, displacement, and geometric linearity of the subgrade adjacent to the existing line during the construction of the pile-raft foundation reinforcement. This research is helpful to supplement the research gaps in the construction of high-speed railway near the existing line and provides a scientific basis for the safe operation of the existing line during the construction of the pile-raft foundation reinforcement.

## 2. Analysis of Construction Impact on Existing Lines

The area adjacent to the parallel section of the Beijing–Shanghai and Shanghai–Nanjing lines was selected as the on-site test section. The test section is located at DK95 + 650 of the newly built Shanghai–Nanjing Intercity Railway (corresponding to K1243 + 100 of the existing Beijing–Shanghai line), and the old and new lines share a drain groove, as shown in Figure 1. The subgrade of the test section is covered with silty clay ranging from 2 to 6 m.

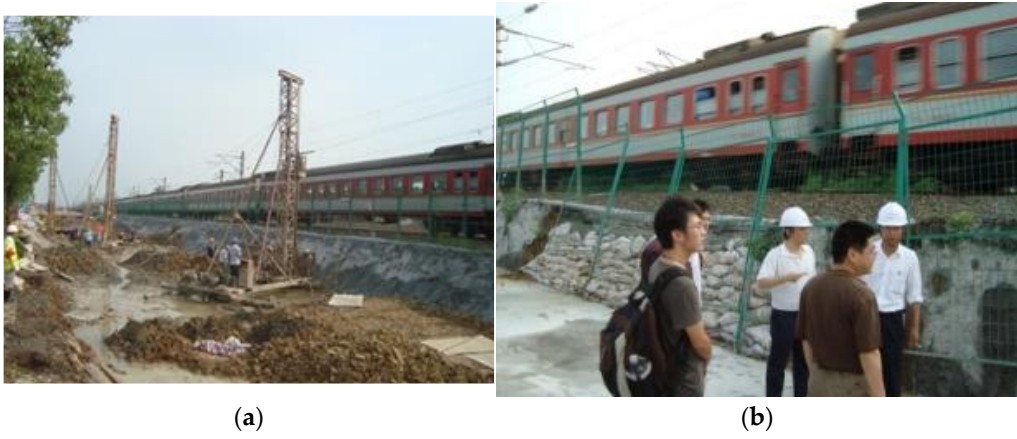

| (**a**) | (**b**) |

**Figure 1.** Engineering field. (**a**) Subgrade pit; (**b**) adjacent existing lines.

Its strata are as follows: (1) $Q_4^{al + pl}$ (Quaternary Holocene alluvial–pluvial layer): silty clay; brownish gray; hard to soft plastic; thickness 0–2 m, $\sigma_0 = 120$ kPa; (2) $Q_4^{al + pl}$ (Holocene alluvial–pluvial): muddy silty clay; gray, soft plastic; thickness 0~11.6 m, $\sigma_0 = 80$ kPa; (3) $Q_3^{al}$ (Fourth series update system): silty clay; brownish yellow, hard plastic; thickness 1.1~29.1 m, $\sigma_0 = 200$ kPa; (4) granodiorite, light yellow and gray, fully weathered. The specific physical and mechanical properties of the soil layer are shown in Table 1.

**Table 1.** The physical and mechanical properties of the soil.

| Soil Layer Name | Formation Thickness/m | Moisture Capacity $\omega$/% | Density $\rho$/g/cm$^3$ | Void Ratio e | Liquid Limit $\omega_L$/% | Plastic Limit $\omega_p$/% | Plasticity Index $I_P$ | Liquidity Index $I_L$ | $a_v$/MPa$^{-1}$ | Constrained Modulus $E_s$/MPa | Cohesion $C_u$/kPa | Internal Friction Angle $\varphi_u$/° | Specific Penetration Resistance $P_sMPa$ |
|---|---|---|---|---|---|---|---|---|---|---|---|---|---|
| (1) Silty clay | 0.6 | 25.9 | 1.99 | 0.72 | 31.3 | 19.3 | 11.97 | 0.55 | 0.29 | 5.97 | 27.0 | 11.8 | |
| (2) Muddy silty clay | 8.8 | 35.0 | 1.90 | 0.99 | 32.4 | 19.7 | 12.64 | 0.82 | 0.41 | 4.68 | 16.0 | 7.6 | 0.39 |
| (3) Silty clay | 11.7 | 23.1 | 2.02 | 0.66 | 32.7 | 19.7 | 13.04 | 0.27 | 0.14 | 12.8 | 29.2 | 12.5 | 1.8 |
| (4) Full regolith | 11.05 | | | | | | | | | | | | |

In order to meet the post-construction settlement control requirements for the high-speed railway, the Shanghai–Nanjing Passenger Dedicated Line adopted a new type of pile-raft foundation. The foundation reinforcement at the adjacent positions of the two lines uses grouted a gravel pile, and the pile top is a raft structure.

The existing Beijing–Shanghai line was built in the 1970s, and the foundation has essentially been consolidated after many years of operation. During the construction of the new Shanghai–Nanjing Passenger Dedicated Line, a shallow foundation pit was excavated on the side of the Beijing–Shanghai subgrade, which changed the existing line stress field and displacement field; the release of stress had an inevitable impact on the output of the existing subgrade. In addition, the soil-squeezing effect during the pile-forming process and the additional stress generated by the filling of the new line foundation caused different degrees of disturbance to the existing line foundation. The process of the influence is shown in Figure 2.

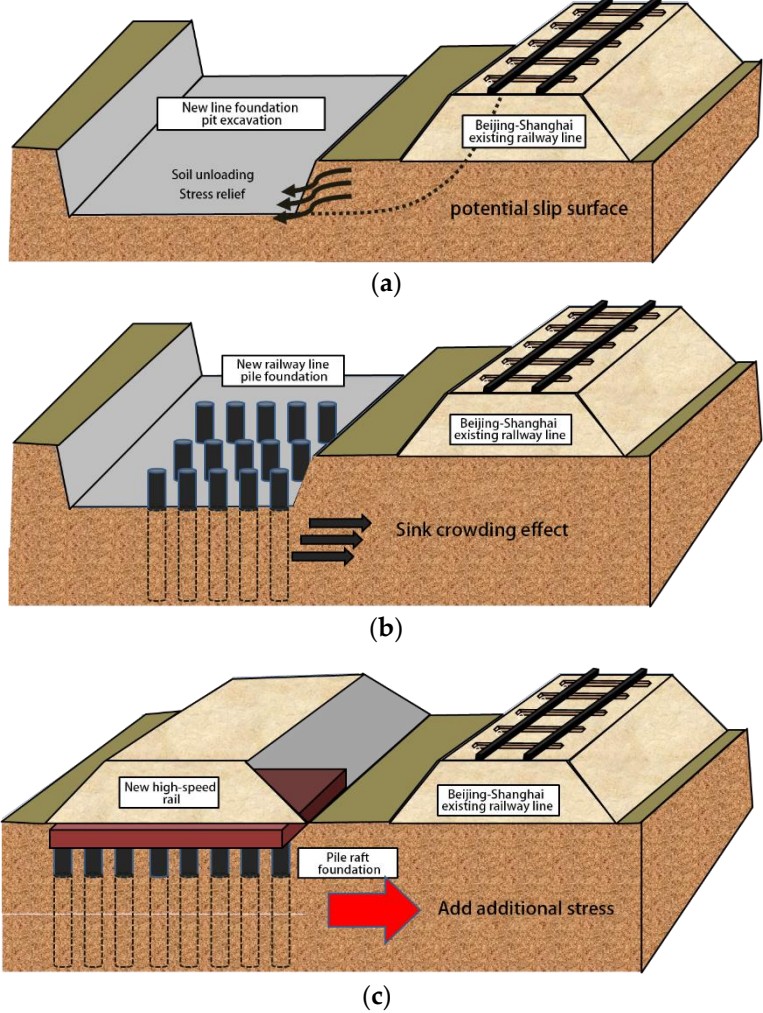

**Figure 2.** Diagram of the influence of the new line construction on the existing subgrade. (**a**) Impact of foundation excavation on existing lines; (**b**) impact of piling on existing lines; (**c**) impact of roadbed filling on existing lines.

## 3. Testing on the Spot

The existing roadbed monitoring tests were carried out on the project site. The tests mainly measured the lateral displacement of the soil between new and old lines, the horizontal displacement of the excavated roadbed slope toe, and the horizontal stress of soil layers at different depths.

In order to observe the lateral displacement of the deep soil, two inclinometer pipes were buried on site. The inclinometer pipes were buried at a depth of 22 m and placed at the foot of the excavation slope of the existing roadbed (the edge of the raft plate of the new pile-raft foundation). In order to measure the horizontal displacement of the slope toe of the existing line base more accurately, the horizontal soil-strain gauges were buried at depths of 0.5, 0.8, and 1.5 m at the position of the excavation slope toe, and one was placed every 1.8 m along the longitudinal direction of the line. The embedded horizontal strain gauge used in the test was less affected by construction interference and had high sensitivity. The element was embedded horizontally in the soil, and the preset shrinkage amount was 20 cm (that is, the test limit); after burying it, the surrounding area was covered with medium-coarse sand and compacted, the lead wire was protected by a wire hose, and the internal spring of the strain gauge was under the action of the horizontal load. Displacement was generated, and the displacement data were output through the electric frequency with centimeter accuracy and were used for the high-precision testing of the lateral displacement of the existing line foundation slope toe during construction.

In order to measure the change in the stress-field distribution between the old and new subgrades during the construction period, stress shovels were buried to test the horizontal lateral stress at different depths between the two lines. The stress shovels were divided into two groups, each of which contained five components with burial depths of 1, 3, 5, 7, and 9 m; the two groups of elements faced the new Shanghai–Nanjing line and the existing Beijing–Shanghai line, respectively. The test point components were buried as shown in Figures 3 and 4.

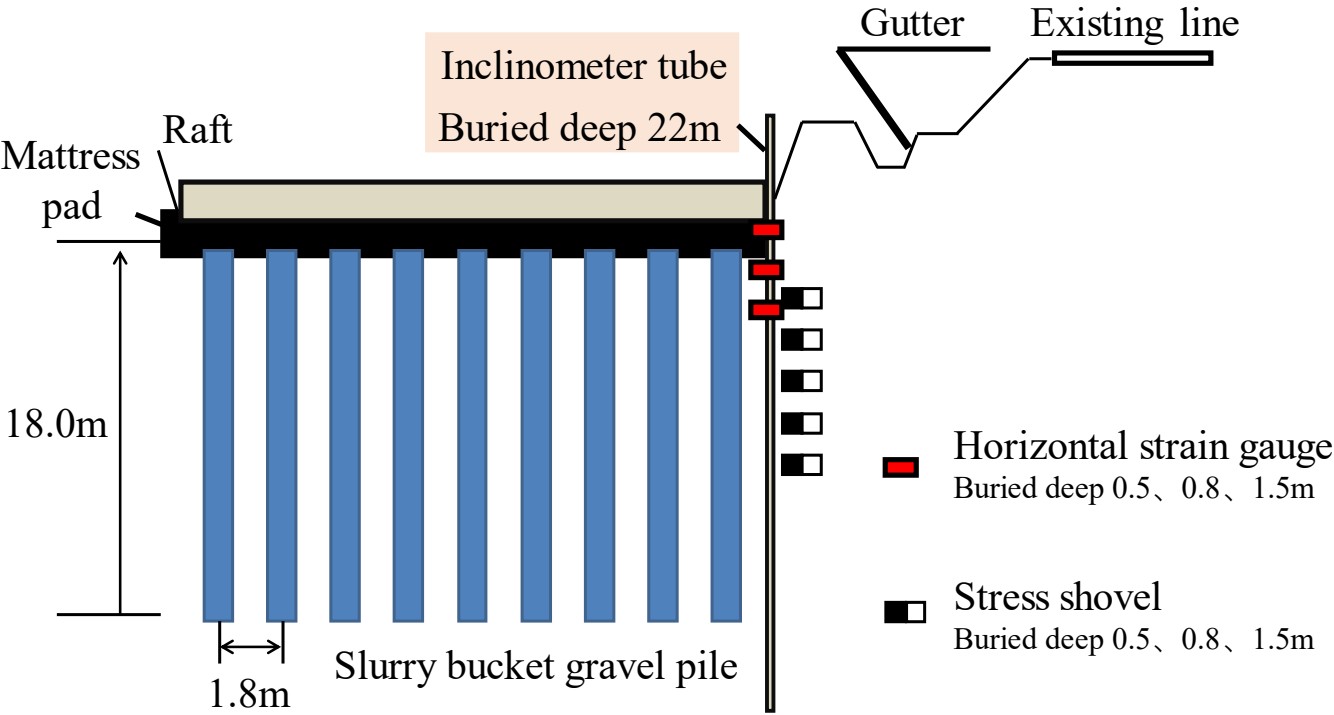

**Figure 3.** The burial profiles of the test components.

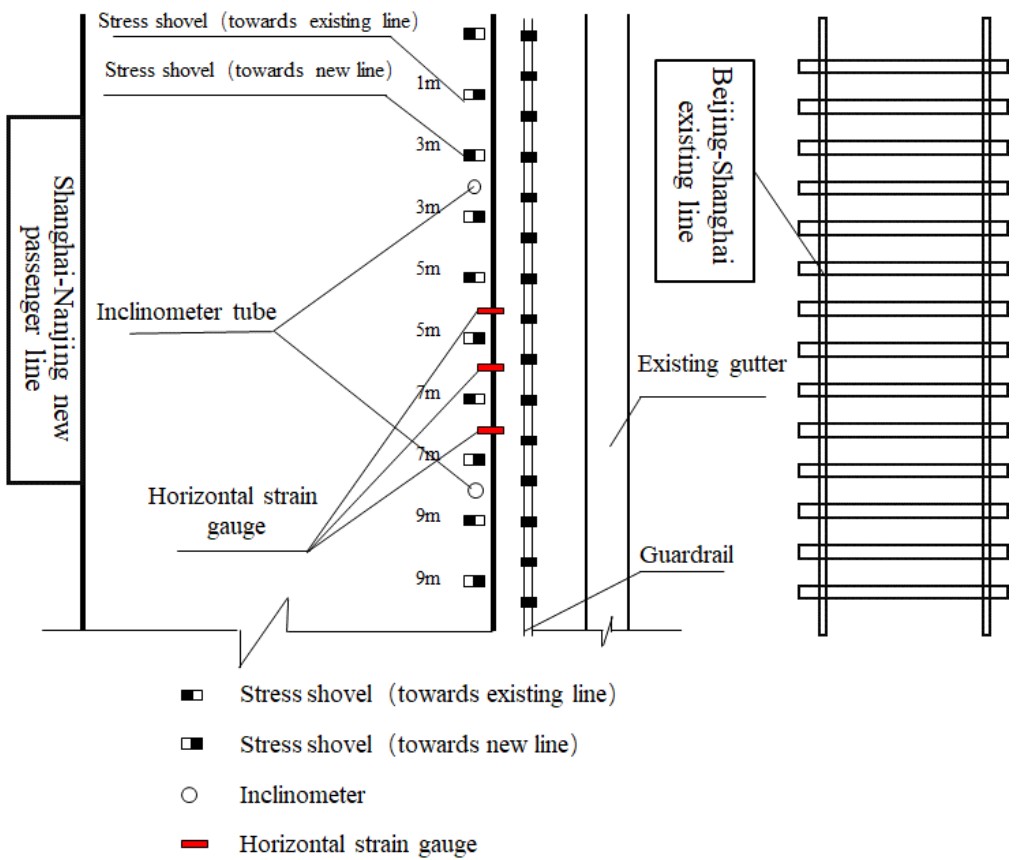

**Figure 4.** Test component layout diagram.

## 4. Analysis of Test Results

### 4.1. Horizontal Displacement of Deep Foundation Soil between Two Lines

The inclinometer tube was used to obtain the change in the lateral displacement distribution in the deep foundation soil. The test showed that the maximum lateral displacement of the foundation soil occurred on the surface during construction, which was the lateral displacement of the new subgrade toward the existing subgrade after the subgrade-filling stage was completed.

The change rule can be seen in Figure 5: (1) During the excavation stage, the deep foundation soil was displaced to the outside (taking the existing roadbed as the starting point), which was caused by the stress release from the unloading of the excavated soil of the new roadbed foundation pit; the maximum lateral displacement was 20.59 mm, and the displacement rate was 0.32 mm/d. However, because of the construction disturbance on the ground surface, the maximum value did not appear near the ground surface, but appeared at a depth of about 10 m. We concluded that this was related to the properties of the foundation soil. (2) During the pile-forming stage, under the influence of the soil-squeezing effect of the pile body, lateral deformation of the foundation soil to the existing line foundation occurred. (3) During the subgrade-filling stage, the lateral displacement continued to increase. For these stages, the data were retracted in October, and the reason for this analysis was the unloading of the pre-compressed soil of the new roadbed. During the construction period, the average displacement rate in the test was 0.59 mm/d, meeting the requirements for deformation control during the construction of new roadbed filling.

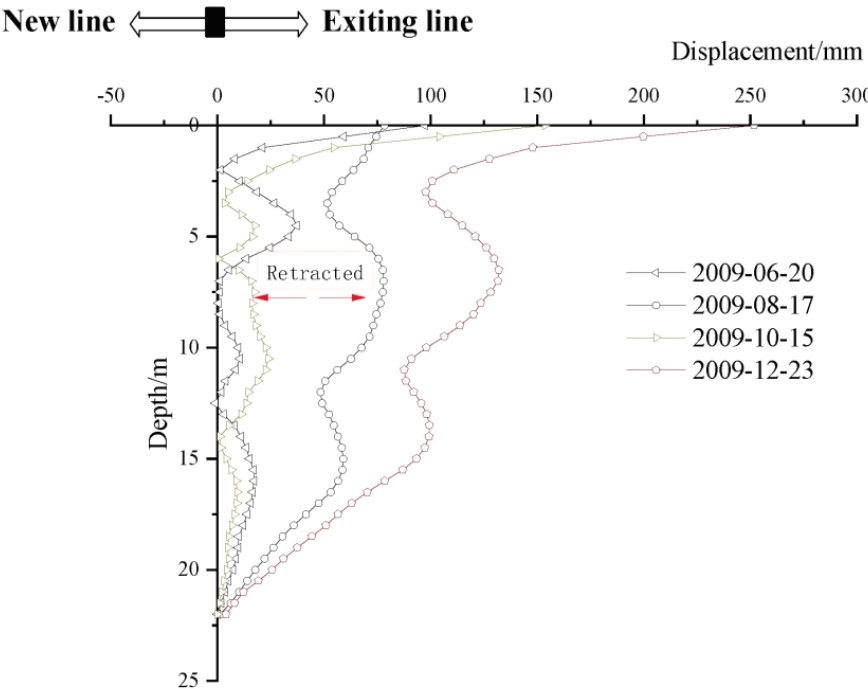

**Figure 5.** Horizontal displacement time curves for deep foundation soil.

### 4.2. Horizontal Displacement of Existing Line Foundation Slope Toe

Figure 6 shows the load changes for the newly built subgrade during the test phase. Later displacements and stress changes corresponded to this loading date. The test results for the inclinometer tube showed that the excavation stage of the foundation pit was affected by the unloading of the soil, and the foundation soil of the existing road base moved out, but the surface data were greatly disturbed by the construction, so a horizontal soil-strain gauge was used to test the horizontal displacement of the foundation slope of the existing line foot. The horizontal displacement of the existing line foundation slope toe and the displacement curve are shown in Figure 7. It can be seen that the displacement change rule at this point was that the deformation developed most rapidly in the excavation stage of the foundation pit, and a slight rebound occurred during the raft-pouring period. We believe that the raft had a strengthening effect on the existing subgrade slope. The displacement development was relatively gentle, and there was a significant increase again after the preload was unloaded.

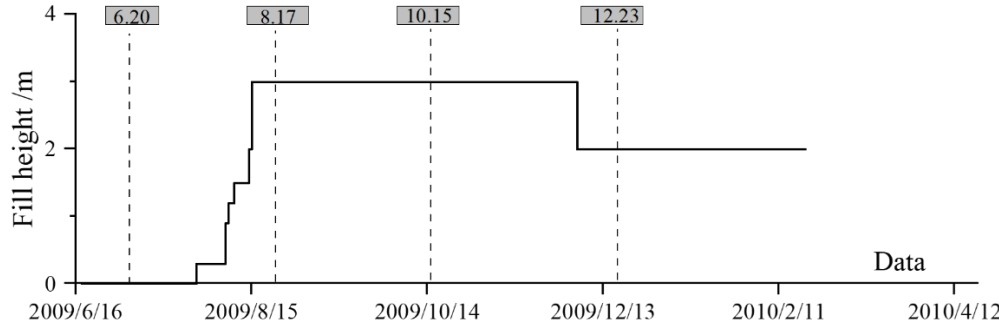

**Figure 6.** Timeline of new subgrade filling.

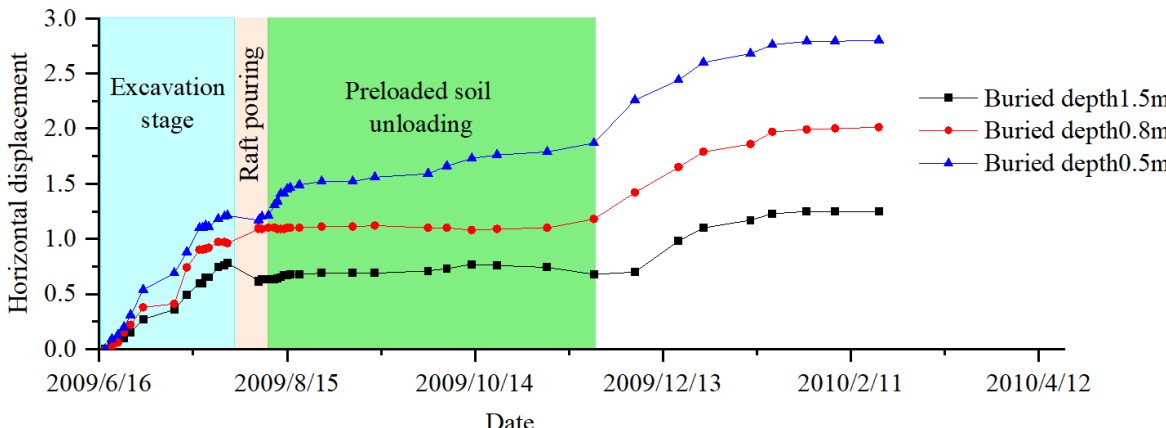

**Figure 7.** Change curves for relative horizontal displacement.

Considering the horizontal displacement of the fixed end of the element during construction, the test result was equivalent to the relative displacement of the soil within the length of the element (40 cm). By combining it with the finite-element estimation, the actual displacement of the existing line foundation slope toe [17] was calculated, and the calculation results are as follows: Figure 8 shows the horizontal displacement of the excavated slope toe node and the node at a horizontal distance of 0.5 m from the slope toe. The finite-element calculation showed that the excavation and unloading of the slope toe had obvious camber displacement, which occurred with the simulated raft and roadbed filling. The displacement was suppressed, or there was even retraction.

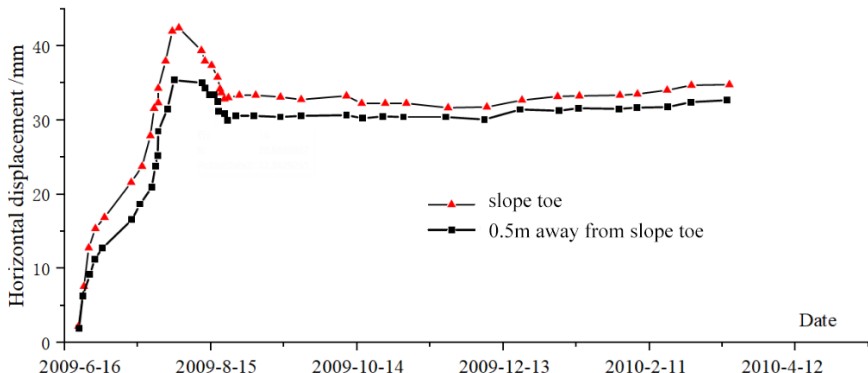

**Figure 8.** Finite-element slope toe horizontal displacement.

The distance between the node at the slope toe of the finite-element model and the node at a distance of 0.5 m was regarded as the test range of the soil-strain gauge; the displacement difference, $H_2$, between the two points was counted, and the ratio of the measured result, $H_1$ (0.5 m-deep soil-strain gauge test data), to the calculated result, $H_2$, was used as the conversion coefficient, a. Owing to the large difference in the rates of change in displacement in different stages, the conversion coefficient, a, was divided into three distinct stages, as shown in Figure 9. The distribution of the conversion coefficient, a, was obtained as follows:

$$a = \begin{cases} \in [0.08,\ 0.27],\ \text{average value } a_1 = 0.16,\ (2009-6-18 \text{ to } 2009-8-18) \\ \in [0.47,\ 0.93],\ \text{average value } a_2 = 0.57,\ (2009-8-18 \text{ to } 2009-10-25) \\ \in [1.18,\ 1.49],\ \text{average value } a_3 = 1.46,\ (2009-10-25 \text{ to } 2010-2-18) \end{cases} \quad (1)$$

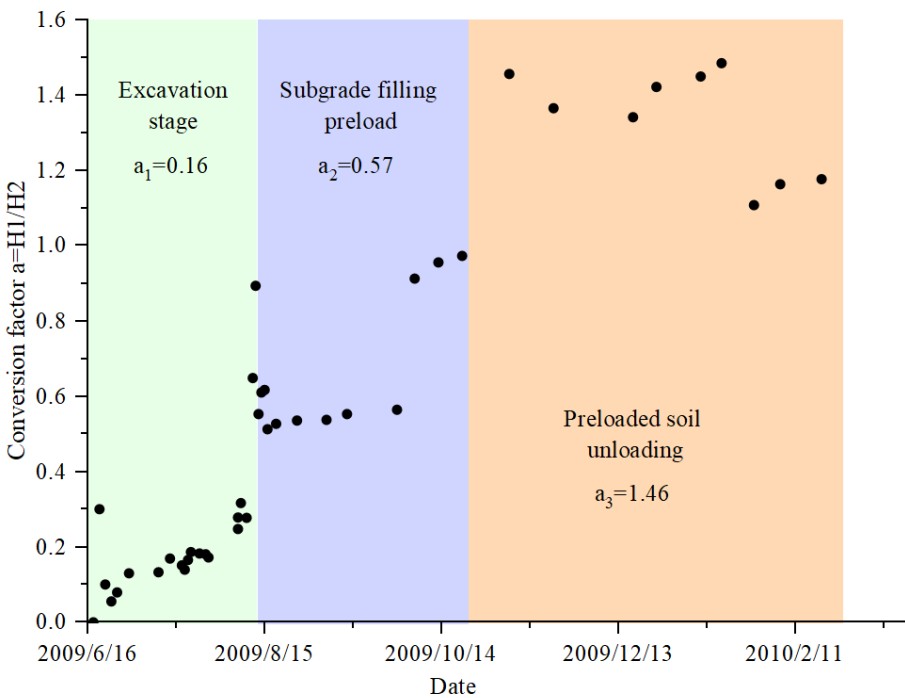

**Figure 9.** Test and calculated conversion coefficient, a.

During the excavation stage of the subgrade and foundation pit (from June to August 2009), the conversion factor was calculated to be $0.08 \leq a \leq 0.27$, and the mean value was $a_1 = 0.16$. When rafts were laid for new roadbeds and the roadbeds were preloaded (August to October 2009), the calculated conversion factor was $0.47 \leq a \leq 0.93$, and the average was $a_2 = 0.57$. In the preloading and unloading stage (October 2009 to February 2010), the conversion factor was calculated to be $1.18 \leq a \leq 1.49$, and the mean value was $a_3 = 1.46$. Based on this conversion factor, the finite-element calculation results were revised. After the excavation of the foundation pit, the maximum cumulative horizontal displacement of the slope toe was calculated to be 42.55 mm, and the corresponding conversion factor $a = 0.57$ at this stage, that is, the initial estimated horizontal displacement of the slope toe, was about 24.25 mm.

There is currently no normative basis for monitoring the horizontal displacement and deformation of an existing subgrade. The control of the deformation amount and deformation rate for a new subgrade filling is not applicable to the old subgrade in operation. Considering that the most dangerous stage of the existing subgrade is during the excavation of the foundation pit, the safety level should be the highest at this stage. The recommended alarm values for the first-level foundation pit were as follows: a cumulative horizontal displacement value of 25–60 mm, a relative depth control value for the foundation pit of 0.2–0.7%, and a change rate of 2–10 mm/d. When there is no clear requirement, the maximum horizontal deformation limit for the first-level foundation pit is 0.002 h. The estimated cumulative value of the slope toe displacement was close to the alarm value, so monitoring of the existing roadbed should be strengthened, and necessary protection measures should be taken.

### 4.3. Deep Foundation Horizontal Stress

The stress shovel test can obtain the variation law for the horizontal earth pressure of the deep foundation between two subgrades, as shown in Figures 10 and 11. The following is known: (1) During the excavation stage, the existing line foundation forms a slope, and the soil gradually changes from exhibiting static earth pressure to exhibiting active earth pressure. The test data showed that the stress of the existing line was released, the lateral earth pressure of the old line toward the new line increased significantly, and the

micro-strain of the element reached 6.8 $\varepsilon\mu$ (dimensionless, $\varepsilon\mu = (\Delta L/L) \times 10^{-6}$), which is about 3 kPa. The foundation soil was expressed as the extrusion of the existing roadbed to the new line. (2) In the process of pile formation, the compressive stress generated by the pile body squeezed the soil, causing the lateral earth pressure to fluctuate (reduce) toward the new line. (3) With an increase in the fill load of the new line, the additional stress in the deep soil increased, the existing line foundation squeezed the new line, the lateral earth pressure reversed, the pressure toward the new line decreased sharply, and the new line affected the existing line. The compressive stress of the line increased; in the later stage, with the stability of the fill load, the lateral stress level of the deep foundation did not change significantly.

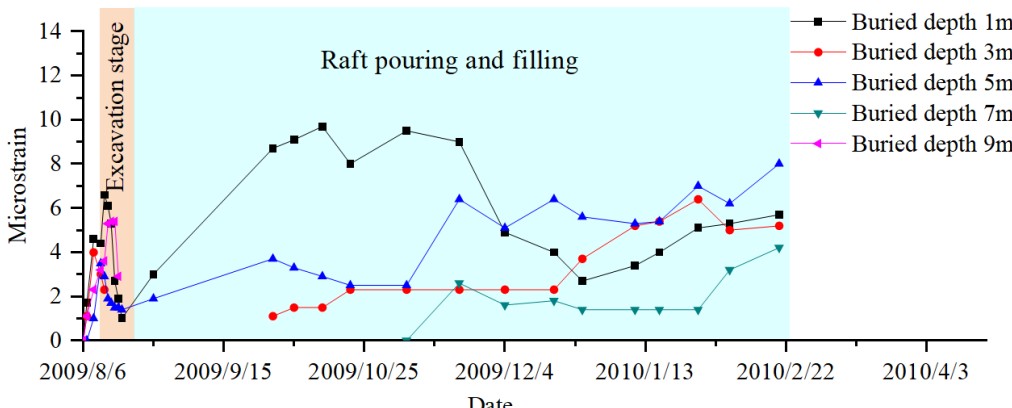

**Figure 10.** Lateral stress curves for existing lines.

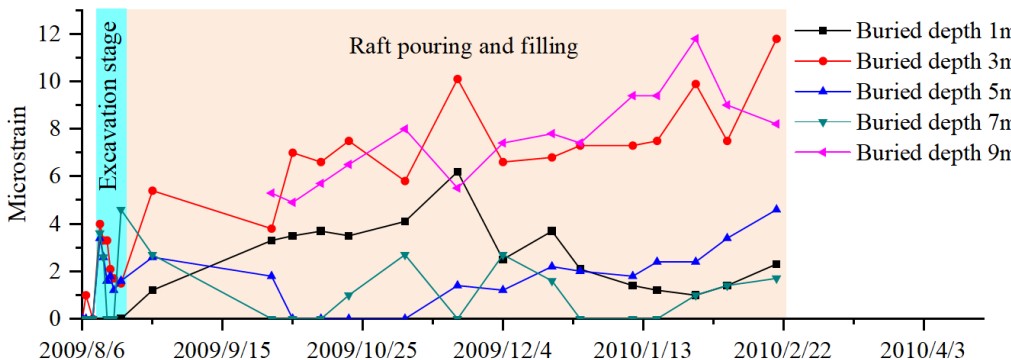

**Figure 11.** Lateral stress curves for new lines.

When further comparing the pressure distribution in the opposite direction, by examining the two-way lateral stress distribution, we can observe in Figure 12 that the extrusion effect of the existing line on the new line was more obvious. This was caused by the load of the trains on the line foundation, which mainly occurred in the shallow foundation. The lateral earth pressure in this test remained less than 10 kPa, which was far less than the highest level (60–70%), *f*, of the alarm value for monitoring foundation pit deformation, and *f* was the designed limit value. The upward pressure did not pose a threat to the existing line base.

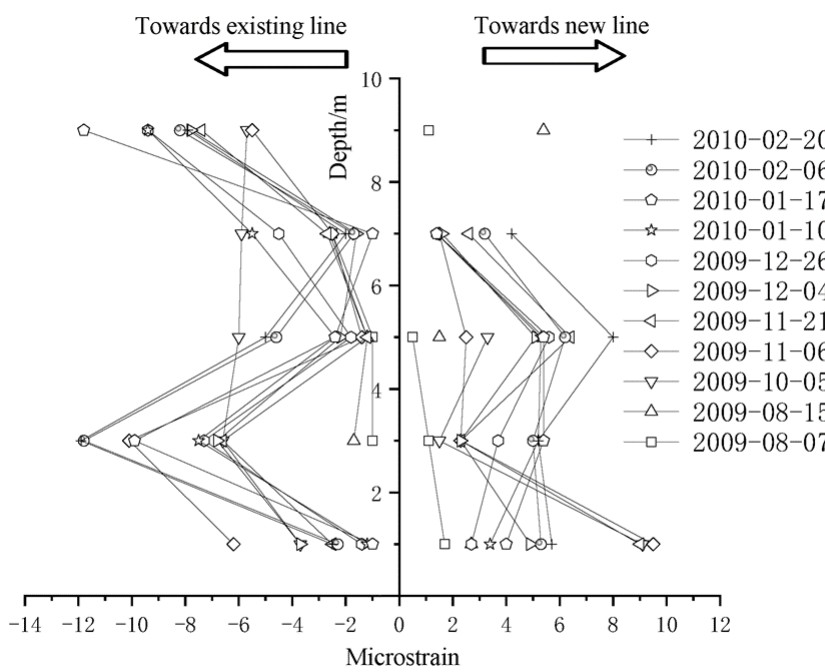

**Figure 12.** Lateral stress distribution curve.

When analyzing the data curves of three indicators such as horizontal stress, lateral displacement, and displacement of existing line foundation slope toe, it can be seen that the most obvious data change occurred in the stage of foundation excavation during the construction process near the existing line. The change trend slows down in the stage of pile formation and paving. The results show that the excavation and unloading of the foundation soil has the greatest impact on the stress field and displacement field distribution of the existing line foundation, and the dynamic stress of the accessories generated by the train load aggravates the trend. The deformation observation of the existing line base should be strengthened to ensure the stability of the embankment, because the existing road foundation is in the most unfavorable state at this stage.

## 5. Stability Evaluation for Existing Line Foundation during the Construction Period

### 5.1. Indicator Selection and Determination

The test results were consistent with previous results from laboratory model tests. In a study of the failure mode of a reinforced soil steep slope [18,19], it was found that a slope with a safety factor for slope failure greater than 1.0 collapsed, while a slope with a safety factor of less than 1.0 remained stable. The quantitative index was screened by using the model test, and the lateral displacement was determined as the quantitative index of slope stability using the fuzzy mathematical analysis method. Based on the above analysis, the horizontal displacement of the subgrade toe was selected as an important control index for the stability of the existing subgrade. The lateral displacement of the deep foundation mainly reflected the deformation of the newly built subgrade, and the deep foundation stress mainly reflected the stress distribution state of the new high-speed railway and the existing railway. These two indicators helped to reveal the variation law of stress and deformation in the foundation during the construction of the pile-raft foundation but could not directly reflect the state of the existing subgrade.

### 5.2. Track-Inspection Car Verification

The current assessment of the condition of an operating railway in China was mainly based on a combination of the condition of the section track quality and local unevenness management, a scoring method for obtaining a deduction value per kilometer, and the standard deviation method for obtaining the track quality index (TQI) value for different

sections. Both methods required a track-inspection vehicle for their implementation. The scoring method is simpler but does not reflect the rate of change in track unevenness, the length of overruns, or periodic continuous unevenness. The standard deviation algorithm usually determines the TQI to be 200 m for a section, taking into account the gauge, height, level, alignment, and track twist; on seven items in the standard deviation calculation, the TQI exceeded the limit value that should trigger maintenance and repair. For the Beijing–Shanghai Railway, it was determined that a TQI greater than 15 is over the limit, and the system must be maintained and repaired if this is exceeded, as shown in Table 2.

**Table 2.** Railway management standard.

| Project | | High | Alignment | Gauge | Level | Track Twist | TQI |
|---|---|---|---|---|---|---|---|
| Management value | $V_{max} < 160$ km/h | $2.5 \times 2$ | $2.2 \times 2$ | 1.6 | 1.9 | 2.1 | 15.0 |
| | $V_{max} > 160$ km/h | $1.5 \times 2$ | $1.6 \times 2$ | 1.1 | 1.3 | 1.4 | 10.0 |

The foundation excavation and piling stage of the project was completed in July, and the roadbed filling was carried out in August. Therefore, this period was selected to test the location of the section (Beijing–Shanghai K1243 + 100) of existing railway waveform data for analysis. The analysis used the scoring method and standard deviation method, and the results of the scoring method are shown in Figure 13, indicating that the deduction of points during the construction phase (July and August) did not differ from that in other normal periods.

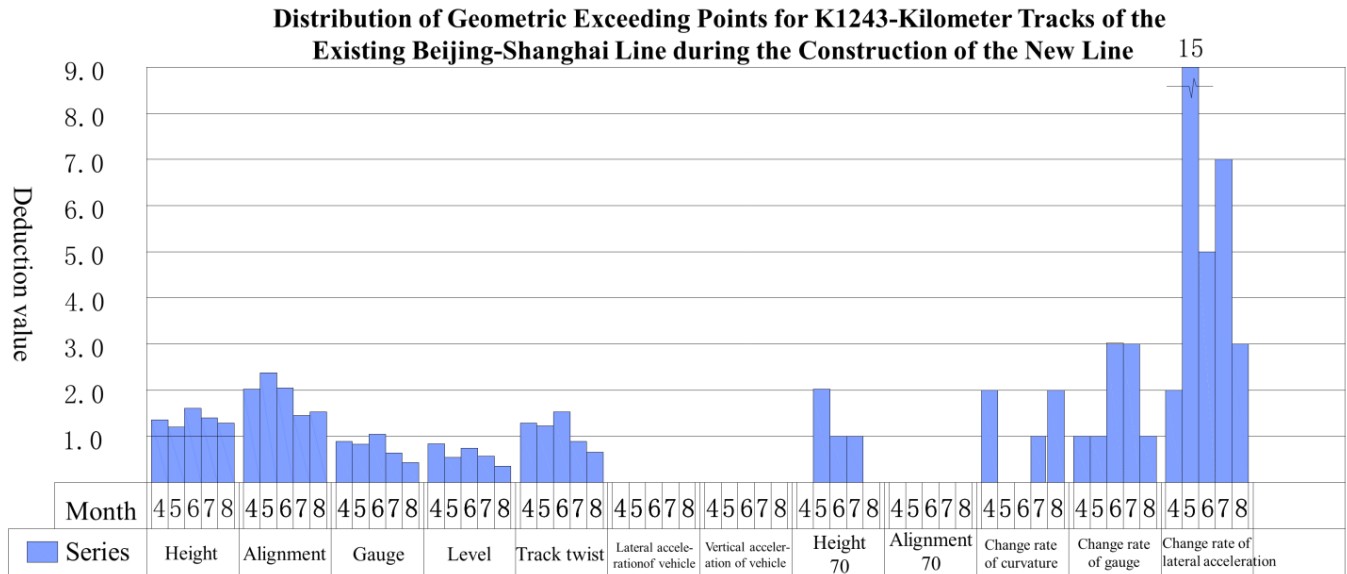

**Figure 13.** Track geometrical transfinite single distribution per kilometer.

The TQI value obtained by the standard deviation method is shown in Figure 14. It can be seen that the TQI value increased significantly after the beginning of August, from 4.26 on 22 May to 9.78 on 22 August, an increase of 129.58%, indicating the obvious influence of construction on the existing lines. However, the standard management value was not exceeded, and maintenance and repair were not required, so the monitoring data were used as the basis for the existing line safety analysis.

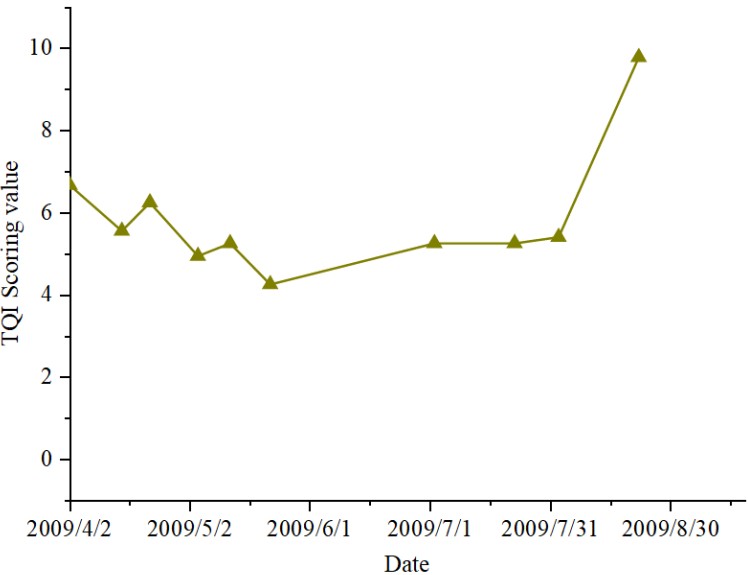

**Figure 14.** Standard deviation method results showing value-change curves for 200 m TQI values.

From the inspection data of the track-inspection car, the linear change curve of the track during the construction period (April–August) was obtained. According to the orbit quality state TCI and local irregularity data, it was known that the existing line foundation remained in a stable state without excessive deformation during the construction period. Among these, the linear change was the most obvious in August. This period was close to the foundation excavation stage, which was mutually verified with the experimental analysis results. The existing line foundations adjacent to the foundation excavation stage are most affected and are in the most unfavorable state. Field test detection and track-inspection car detection data can complement and verify each other. The former obtains the internal deformation data of the subgrade, and the latter obtains the track deformation data of the subgrade superstructure to jointly ensure the operation safety of the existing subgrade during construction.

## 6. Conclusions

In order to ensure the construction quality of a high-speed-rail pile-raft foundation and the operation safety of the adjacent existing railway, this study conducted on-site tests and research on engineering protection measures. The research focused on three stages of construction—foundation excavation, foundation treatment, and roadbed filling—using soil-strain gauges (buried in the foot of the existing roadbed slope), stress shovels (buried in the deep foundation facing the existing line and the new line), and inclination measurements. A deformation stress test was carried out on the pipe (buried in the foundation between the old and new lines), and the following conclusions were obtained:

1. Among the three stages of high-speed rail pile-raft foundation construction adjacent to the existing line, the stress and deformation changes monitored were the most obvious during the foundation excavation stage. This stage was the most unfavorable state of the adjacent existing roadbed. The safety monitoring of the existing roadbed should be strengthened at this stage.

2. The results of the track-inspection car showed that this index should be used as the monitoring and control index for the stability of the subgrade slope during the construction period, because the horizontal displacement of the existing line foundation slope toe is the most sensitive to construction disturbance and has regularity.

3. Monitoring of existing line foundation stress, horizontal displacement of the slope toe, and lateral displacement of deep foundation during construction. This work can effectively monitor the internal stress and deformation of the subgrade of the existing

line. The data analysis of the track-inspection car can realize the linear deformation detection of the existing railway track. These two works complemented each other. This work can realize the internal and external safety monitoring of the subgrade to ensure the safe operation of the existing subgrade.

**Author Contributions:** Conceptualization, S.Z. and P.L.; methodology, S.Z. and X.Z.; software, S.Z.; validation, X.Z., J.L. and X.C.; formal analysis, X.Z.; investigation, S.Z. and X.Z.; resources, S.Z.; data curation, J.L.; writing—original draft preparation, P.L.; writing—review and editing, P.L.; visualization, S.Z.; supervision, X.Z.; project administration, S.Z. and J.L.; funding acquisition, S.Z. All authors have read and agreed to the published version of the manuscript.

**Funding:** The National Natural Science Foundation of China (No. 51208517, No. 52178182, and No. u1134207); The National Science Foundation for Post-doctoral Scientists of China (No. 2013m530360) and the China Railway Science and Technology Research and Development Plan Project (grant numbers 2020-Major project-02, 2021-Major project-02, and 2021-Key projects-11); National Science Foundation for Distinguished Young Scholars of Hunan Province of China (2022JJ10075).

**Institutional Review Board Statement:** Not applicable.

**Informed Consent Statement:** Not applicable.

**Data Availability Statement:** Not applicable.

**Conflicts of Interest:** The authors declare no conflict of interest.

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
