# Peer review of "Analysis of the Influence of Pile-Raft Foundation Reinforcement on an Adjacent Existing Line Foundation"

_applsci, doi:10.3390/app13052820_

Round 1

Reviewer 1 Report

The effect of construction on the adjacent structures is important and should be investigated in detail, as it is complex. This manuscript presented some on-site monitoring results about the effect of construction on an adjacent existing line foundation, taking the typical sections of the Beijing–Shanghai and Shanghai–Nanjing lines as examples. The results are useful in disaster prevention. However, this manuscript has not been well presented and structured, and thus, a major revision is suggested. Some comments are listed below for improving the quality of the manuscript.

The language should be improved. Although I am not a native speaker, reviewing this manuscript is tired for me, because the sentences are not well written, and the English format is not reading friendly (e.g., lines 112~118). It is suggested to use professional English editing service.

Line 51~53: Reference is missing.

It is suggested to rewrite the abstract, as it is tedious in its current version. Please clarify the issues addressed in this manuscript and the key findings should be highlighted.

Introduction part is also worse, especially about the literature review. The quality should be improved and some newly published works should be included. In addition, the works reviewed in instruction should be better presenting. Perhaps classification would be better.

The most important limitation of this manuscript is lacking the better elucidation of the scientific question. The questions described in this manuscript is too site-specific making the manuscript more like a case study rather than an academic paper. It is suggested to rewrite lines 108~120. Some common questions should be summarized, and thus, the results presented would be more useful and can be used for other cases.

The figure quality should be improved, especially the font, which should be consistent with the text.

Reviewer 2 Report

The manuscript is good and recommended for publication.

Author Response

Thank you for your affirmation of this article

Reviewer 3 Report

Some reasons for rejection:

- The title of the article is not clear with the research purpose. The author pays attention to the content of the abstract, the main part must be related to the title of the article.

- Some of the contents presented in Sections 2, 3, 4 are not clear, mistakenly using experimental results or FEM analysis results.

- Using the test results but not describing the horizontal displacement test equipments below the foundation.

- The positions of measuring ground displacement in depth (Figure 3) do not match the displacement measurement results with Figure 5.

- In addition, some parameters related to track quality index (TQI) are not clearly presented and cited (in section 5).

Round 2

Reviewer 1 Report

The manuscript can be accepted now, and i have no further comments.